# Habitat fragmentation mediates the mechanisms underlying long-term climate-driven thermophilization in birds

**Juan Liu[1], Morgan W Tingley[2], Qiang Wu[1], Peng Ren[1], Tinghao Jin[1], Ping Ding[1]\*, Xingfeng Si[3]\***

[1]MOE Key Laboratory of Biosystems Homeostasis & Protection, College of Life Sciences, Zhejiang University, Hangzhou, China; [2]Department of Ecology and Evolutionary Biology, University of California, Los Angeles, Los Angeles, United States; [3]Center for Global Change and Ecological Forecasting, Zhejiang Zhoushan Island Ecosystem Observation and Research Station, Institute of Eco Chongming, Zhejiang Tiantong Forest Ecosystem National Observation and Research Station, School of Ecological and Environmental Sciences, Shanghai, China

## eLife Assessment

This **fundamental** study substantially advances our understanding of how habitat fragmentation and climate change jointly influence bird community thermophilization in a fragmented island system. The authors provide **convincing** evidence using appropriate and validated methodologies to examine how island area and isolation affect the colonization of warm-adapted species and the extinction of cold-adapted species. This study is of high interest to ecologists and conservation biologists, as it provides insight into how ecosystems and communities respond to climate change.

**\*For correspondence:** dingping@zju.edu.cn (PD); sixf@des.ecnu.edu.cn (XS)

**Competing interest:** The authors declare that no competing interests exist.

**Abstract** Climatic warming can shift community composition driven by the colonization-extinction dynamics of species with different thermal preferences; but simultaneously, habitat fragmentation can mediate species' responses to warming. As this potential interactive effect has proven difficult to test empirically, we collected data on birds over 10 years of climate warming in a reservoir subtropical island system that was formed 65 years ago. We investigated how the mechanisms underlying climate-driven directional change in community composition were mediated by habitat fragmentation. We found thermophilization driven by increasing warm-adapted species and decreasing cold-adapted species in terms of trends in colonization rate, extinction rate, occupancy rate and population size. Critically, colonization rates of warm-adapted species increased faster temporally on smaller or less isolated islands; cold-adapted species generally were lost more quickly temporally on closer islands. This provides support for dispersal limitation and microclimate buffering as primary proxies by which habitat fragmentation mediates species range shift. Overall, this study advances our understanding of biodiversity responses to interacting global change drivers.

## Introduction

One widespread consequence of anthropogenic-caused climate warming is the shift in species' distributions toward cooler regions (*Chen et al., 2011*; *La Sorte and Thompson, 2007*; *Thomas and Lennon, 1999*). As species move across the globe, the composition of community changes, with species both colonizing and departing (*Menéndez et al., 2006*; *Tingley and Beissinger, 2013*); yet

**Figure 1.** The general process of thermophilization and the hypothesized framework of thermophilization in fragmented habitats. Two general processes of thermophilization are shown on the top: increasing colonization rate of warm-adapted species, and increasing extinction rate of cold-adapted species over time (**a**). Compared to continuous habitats, a hotter microclimate caused by lower buffering ability on fragmented patches may attract more warm-adapted species to colonize while causing cold-adapted species to extirpate or emigrate faster (**b**). Loss of cold-adapted species can also be exacerbated when highly fragmented patches harbor lower habitat heterogeneity (e.g. resource, microrefugia), which will also reduce colonization of warm-adapted species (**c**). Isolated patches due to habitat fragmentation will block warm-adapted species' colonization and cold-adapted species' emigration under warming (**d**). The three distinct patches signify a fragmented landscape and the community in the middle of the three patches was selected to exemplify colonization-extinction dynamics in fragmented habitats. Relative species richness is shown by the number of bird silhouettes in the community . Note that extinction here may include both the emigration of species and then the local extinction of species and that CTI can also change simply due to differential colonization-extinction rates by thermal affinity if the system is not at equilibrium prior to the study. In our study system, we have no way of knowing whether our island system was at equilibrium at onset of the study, thus, focusing on changing rates of colonization-extinction over time presents a much stronger tests of thermophilization.

colonizing and departing species are unlikely to be randomly drawn from species pools, but instead will be drawn depending on their traits, such as being warm-adapted versus cold-adapted. As a result, a warming climate should lead to compositional shifts toward relatively more warm-adapted species in local communities (*Thomas et al., 2006*), a phenomenon termed 'thermophilization' (*Feeley et al., 2020*; *Gottfried et al., 2012*; *Zellweger et al., 2020*).

In practice, thermophilization can be measured by a temporal increase in the Community Temperature Index (CTI; *Devictor et al., 2008*) – an estimation of the average temperature experienced by all species in an assemblage throughout their respective ranges. While thermophilization of communities is usually accompanied by an increase in warm-adapted species and a decrease in cold-adapted species (*Curley et al., 2022*; *Princé and Zuckerberg, 2015*; *Tayleur et al., 2016*; *Figure 1a*), this process is usually spatially heterogeneous and depends on the availability of current habitat and the ability to disperse into newly suitable habitats (*Gaüzère et al., 2017*; *Richard et al., 2021*), both of which will be directly or indirectly affected by other global change stressors. Habitat fragmentation, usually defined as the process of transforming continuous habitat into spatially isolated and small patches (*Fahrig, 2003*), in particular, has been hypothesized to have interactive effects with climate change on community dynamics. Among the various ways in which habitat fragmentation is conceptualized and measured, patch area and isolation are two of the most used measures (*Fahrig, 2003*). Specifically, habitat fragmentation usually alters both habitat spatial structure and microclimate characteristics (*Ewers and Banks-Leite, 2013*; *Fahrig, 2003*), either of which can influence thermophilization processes (*Gaüzère et al., 2017*; *Platts et al., 2019*; *Richard et al., 2021*; *Zellweger et al., 2020*). Understanding the interaction between climate change and habitat fragmentation on

community composition is essential to predicting biodiversity outcomes, but we generally lack empirical examples with which to study this phenomenon (*Opdam and Wascher, 2004*).

Theoretically, habitat fragmentation could mediate differential dynamics of warm-adapted versus cold-adapted species either by changing existing habitats' suitability or by limiting dispersal into newly suitable habitats (*Newson et al., 2014*; *Opdam and Wascher, 2004*; *Figure 1*). The overall effect of fragmentation on climatic niche tracking of communities can be decomposed into three different mechanisms, each of which has been proposed and supported with different degrees of evidence. First, fragmented habitats may more weakly buffer macroclimate than intact habitats (*Ewers and Banks-Leite, 2013*), resulting in fragmented patches experiencing relatively higher temperatures. Increased temperature, especially in small patches, may boost the colonization or population growth rate of warm-adapted species but cause higher extinction risk or demographic declines of cold-adapted species (*Figure 1b*). This microclimate mechanism has been proposed as an important factor driving thermophilization in plant communities (*Zellweger et al., 2020*).

Second, larger patches generally contain higher habitat heterogeneity and therefore also encompass higher food availability and microrefugia (*Liu et al., 2020*). A diversity of habitats and resources is closely related to a species' successful establishment and can act as a buffer against extinction under climate change (*Luoto and Heikkinen, 2008*). For instance, higher habitat diversity can boost the ability of niche tracking in bird communities (*Gaüzère et al., 2017*), mainly by fostering species' dispersal into new, climatically suitable sites. Similarly, in areas where topography generates greater microclimate variation, population losses of plants and insects due to climate change are substantially reduced (*Suggitt et al., 2018*). We thus expect that the reduction in habitat heterogeneity associated with higher fragmentation will intensify the loss of cold-adapted species and weaken the colonization of warm-adapted species (*Figure 1c*).

Third, habitat fragmentation could prevent or interrupt species' dispersal which is essential for them to track suitable climatic niches (*McGuire et al., 2016*; *Opdam and Wascher, 2004*; *Thomas et al., 2006*). This limitation may come from higher mortality rate with increasing dispersal distance during searching for suitable patches (*Brooker et al., 1999*), and will influence the colonization-extinction dynamics underlying climate response (*Fourcade et al., 2021*; *Hylander and Ehrlén, 2013*). For example, habitat fragmentation renders habitats to be too isolated to be colonized, causing sedentary butterflies to lag more behind climate warming in Britain than mobile ones (*Warren et al., 2001*). All things being equal, less isolated patches should experience increasing rate in colonization of warm-adapted species under warming, benefitting from near sources of species; meanwhile, cold-adapted species should experience higher extinction rates on less isolated patches because they can emigrate easily from these patches under climate warming (*Figure 1d*).

Although the overall effect of habitat fragmentation on climate-induced community dynamics has been established, few empirical studies have ever considered these multiple mechanisms simultaneously. Here, we use 10 years of bird community data in a subtropical land-bridge island system (Thousand Island Lake, TIL, China, *Figure 2*) during a period of consistent climatic warming (*Figure 2— figure supplement 1*) to explore whether and how community thermal niche composition responds to climate change and how habitat fragmentation mediates the process. This reservoir island system was formed 65 years ago due to dam construction. Specifically, we focused on the following three predictions:

1. We predict the existence of thermophilization of bird communities in our fragmented land-bridge island system, characterized by an increasing temporal trend of CTI.
2. We predict that thermophilization of bird communities should result from both increasing warm-adapted species and abundances and decreasing cold-adapted species and abundances. We also predict that, over time with a warming climate, warm-adapted species should increase their colonization rate while cold-adapted species should increase their extinction rate (*Figure 1a*).
3. We predict that habitat fragmentation – measured as island area and distance to the mainland – will mediate the two main driving processes of thermophilization: colonization of warm-adapted species and extinction of cold-adapted species. Following our theoretical framework, if the climate buffering effect dominates (*Figure 1b*) area effects, we expect the differential trends of warm-adapted species versus cold-adapted species to be more extreme on smaller islands (i.e. even faster colonization of warm-adapted, even faster extinction of cold-adapted) due to weaker microclimate buffering. Alternatively, if the effects of habitat heterogeneity dominate (*Figure 1c*) area effects, cold-adapted species will again show greater extinction, but

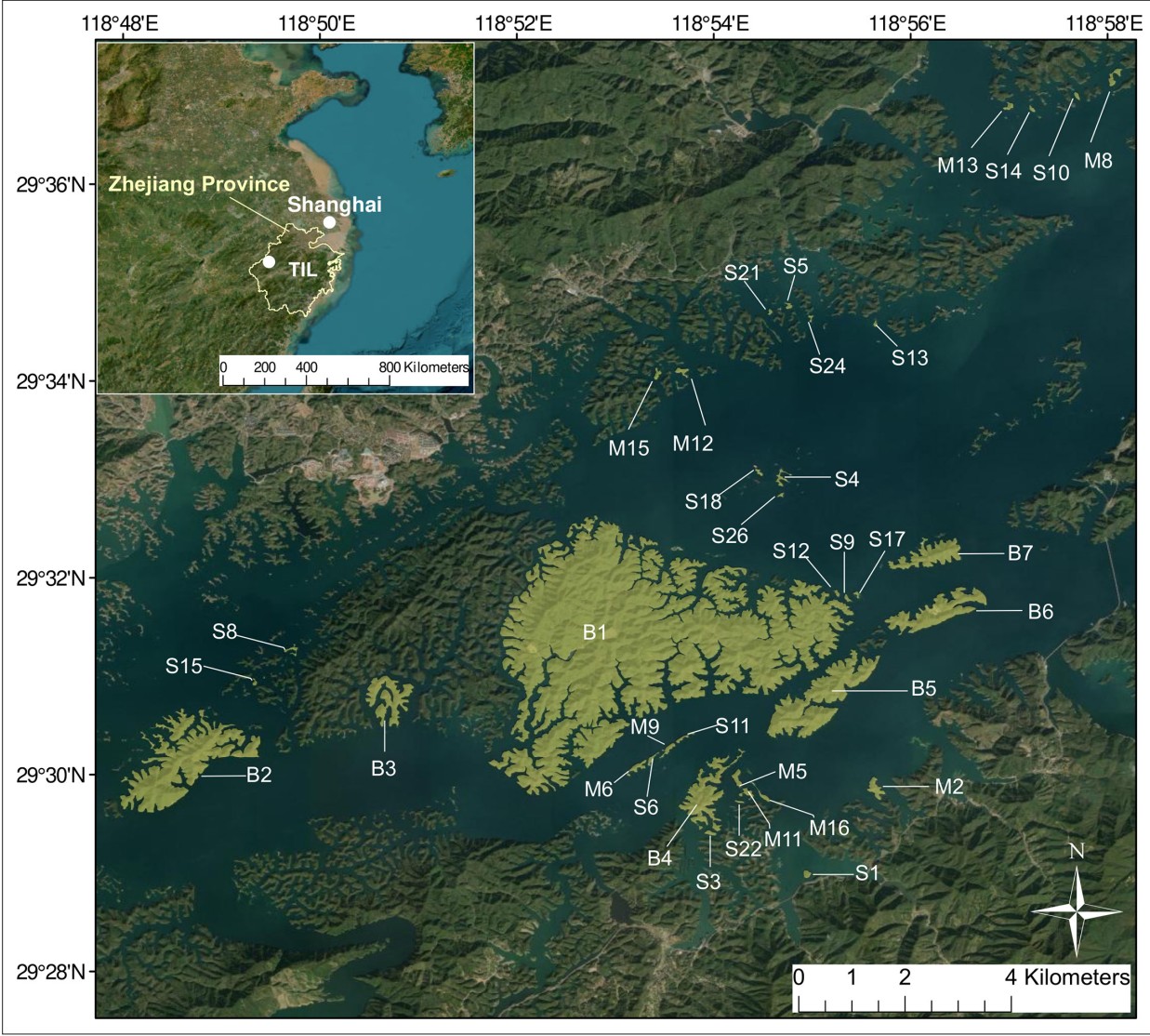

**Figure 2.** The location of Thousand Island Lake and the experimental islands. The map was created using ESRI (Environmental Systems Resource Institute) ArcMap software (version 10.3). The base map sources include Esri, Maxar, Earthstar Geographics, and the GIS User Community.

The online version of this article includes the following figure supplement(s) for figure 2:

**Figure supplement 1.** Breeding season temperature from the year 2012–2021 in the Thousand Island Lake, China.

warm-adapted species will alternatively experience lower colonization on smaller islands (relative to larger islands) due to less abundant resources (*Si et al., 2017*). Finally, we predict that on more isolated islands (*Figure 1d*), the colonization of warm-adapted species and the extinction of cold-adapted species will increase more slowly because isolation hinders both immigration and emigration.

## Results

The number of species detected in surveys on each island across the study period averaged 13.37±6.26 (mean ± SD) species, ranging from 2 to 40 species, with an observed gamma diversity of 60 species. The STI of all 60 birds averaged 19.94±3.58°C (mean ± SD) and ranged from 9.30°C (*Cuculus canorus*) to 27.20°C (*Prinia inornate*), with a median of 20.63°C (*Figure 3—figure supplement 1*; *Figure 3—figure supplement 2*). STI of resident species (n=47) and summer visitors (n=13) did not show a significant difference (t-test: $t$=0.23, df = 17.82, $p$=0.82). No significant correlation was found between STI and species' ecological traits; specifically, the continuous variables of dispersal ability, body size,

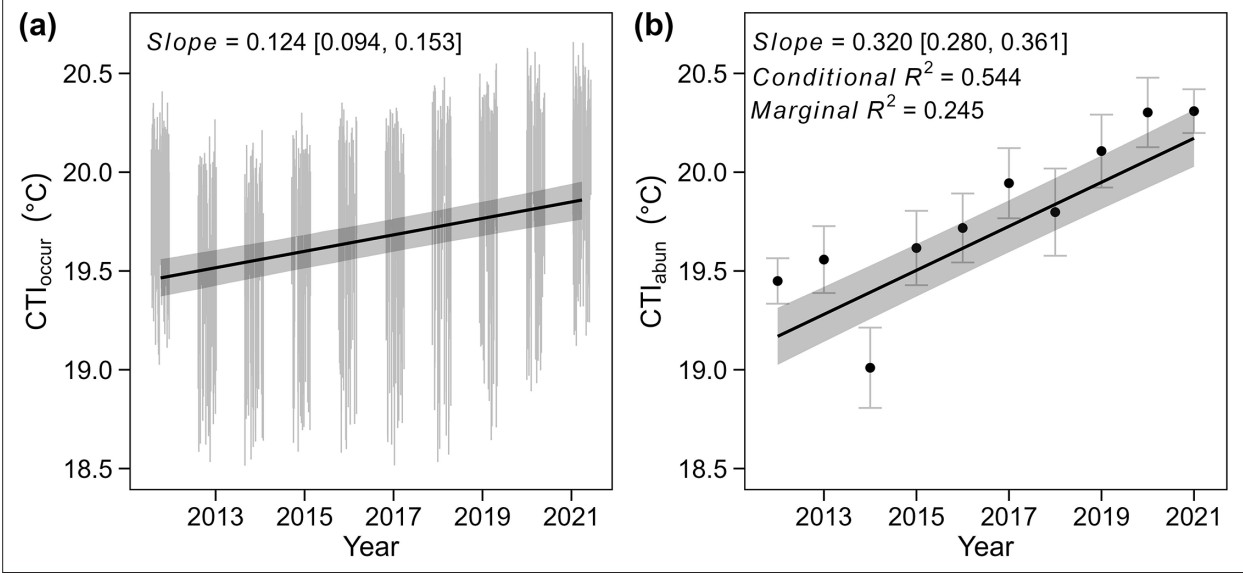

**Figure 3.** Temporal trend in $CTI_{occur}$ and $CTI_{abun}$ in the TIL. Vertical lines in (**a**) are posterior estimated CTI (mean ± SD); solid black line and shaded area are predictions and 95% credible intervals extracted from JAGS modeling posterior mean $CTI_{occur}$ (n = 530) as a function of year, and using island identity as a random effect while accounting for the variation in posterior $CTI_{occur}$. Points and error bars in (**b**) are observed $CTI_{abun}$ (mean ± 2 SD); solid black line and shaded area are predicted values and 95% confidence intervals estimated from LMM modeling observed $CTI_{abun}$ (n = 530) as a function of year, and using island identity as a random effect.

The online version of this article includes the following figure supplement(s) for figure 3:

**Figure supplement 1.** Species temperature index of 60 bird species in the Thousand Islands Lake, China.

**Figure supplement 2.** Robustness of 60 species' STI to changes in data resources.

body mass and clutch size (Pearson correlations for each, $|r|<0.22$), and the categorial variables of diet (carnivorous/omnivorous/herbivory), active layer (canopy/mid/low), and residence type (resident species/summer visitor).

## Thermophilization of bird communities

At the landscape scale, considering species detected across the study area, occurrence-based CTI ($CTI_{occur}$; see section STI and CTI) showed no trend (posterior mean temporal trend = 0.414; 95% CrI: –12.751, 13.554) but abundance-based CTI ($CTI_{abun}$; see section STI and CTI) showed a significant increasing trend (temporal trend [mean ± SE]=0.327 ± 0.041, t=7.989, p<0.001). When measuring CTI trends for individual transects (expressed as °C/unit year), we found significant increases in CTI for both occurrence- (mean temporal trend = 0.124; 95% CrI: 0.108, 0.137) and abundance-based indices (temporal trend = 0.320 ± 0.021, $t_{493}$=15.534, p<0.001; *Figure 3*).

## Mechanisms underlying CTI trends

Comparing across species with different thermal affinities, we found a weak positive linear relationship between STI and species-specific temporal occupancy trends (t-test: slope = 0.206, $t_{58}$=1.766, p=0.082; *Figure 4a*), indicating warm-adapted species were marginally more likely to increase in occurrence over time. Decomposing occurrence change to dynamic parameters; however, we found a significant positive relationship between STI and species' temporal trends in colonization (t-test: slope = 0.259, p=0.043) and a significant negative relationship between STI and species' temporal trends in extinction (t-test: slope = –0.414, p=0.015; *Figure 4—figure supplement 1*). Thus, warm-adapted species generally increased in occupancy over time, which was driven by an increase in colonization probability and a decrease in extinction probability; in contrast, cold-adapted species generally decreased in occupancy over time, which was driven by an increase in extinction probability and a decrease in colonization probability (*Figure 4b*). Considering all species, the influence of a thermal-association gradient underlying occurrence dynamics resulted in a particularly strong

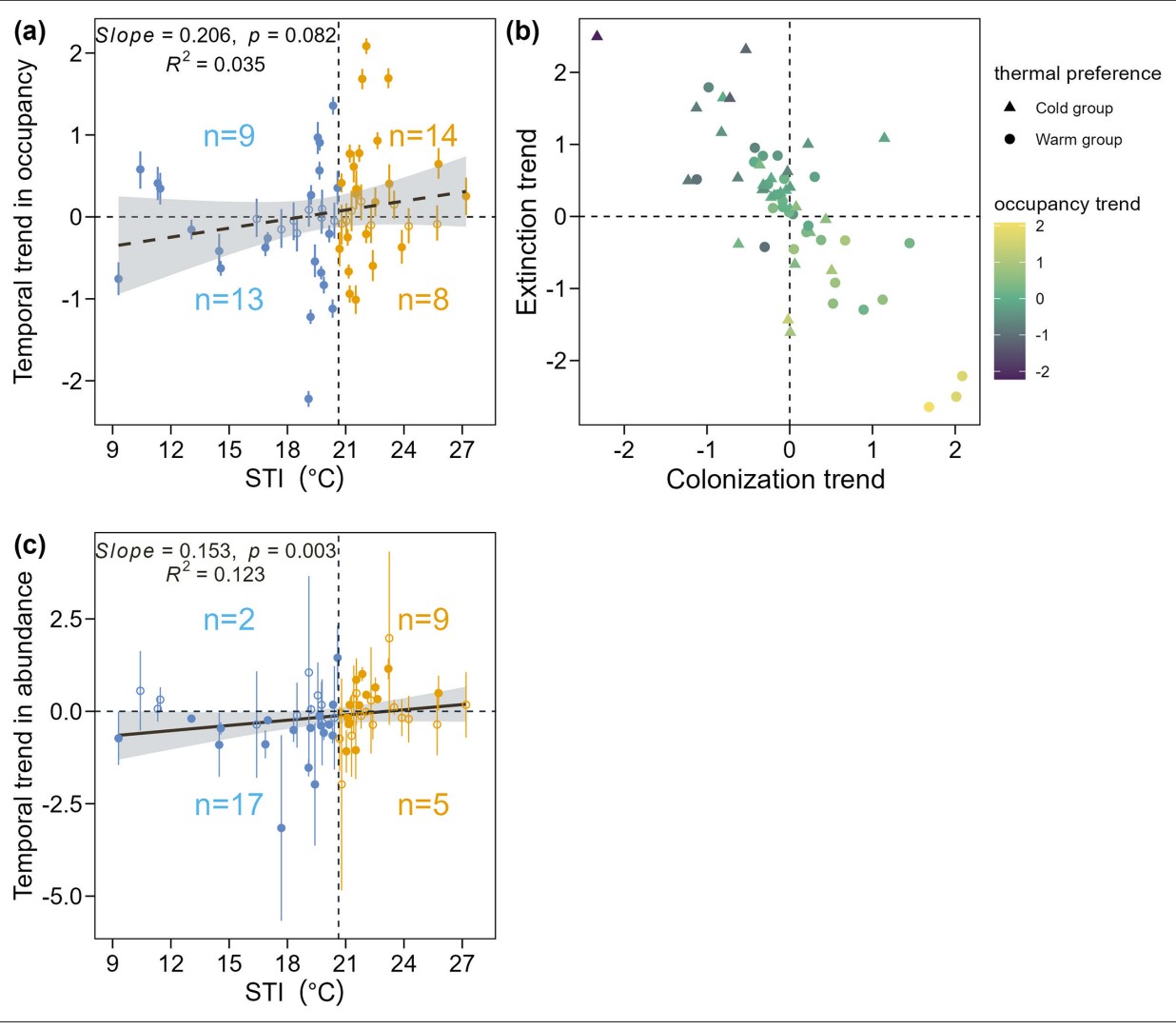

**Figure 4.** Relationship between STI and occupancy trend, abundance trend and the relationship among species thermal preference and trend in occupancy, colonization, and extinction rates. Each point and error bar in (**a**) represents the temporal trend of occupancy rate (posterior mean and 95% credible interval of year effect on occupancy rate) for each species, with the filled dots indicating a significant year effect while the hollow dots indicating nonsignificant year effect. The dotted vertical line indicates the median of STI values. Cold-adapted species are plotted in blue and warm-adapted species are plotted in orange. The number of species with significant occupancy trends in each quadrant was added to the plot. The black dashed line and shaded area are the predicted values and 95% confidence intervals of the weighted linear regression model (n = 60). Each point in (**b**) represents the posterior mean estimate of year in colonization, extinction or occupancy rate for each of 60 species. The color of the point indicates the temporal trend in occupancy. Similar to (**a**), each point and error bar in (**c**) represents the temporal trend of abundance (year effect and 95% confidence interval of year effect on abundance) for each species. The black dashed line and shaded area are the predicted values and 95% confidence intervals of the weighted linear regression model (n = 60).

The online version of this article includes the following figure supplement(s) for figure 4:

**Figure supplement 1.** Relationship between STI and extinction trend or colonization trend.

**Figure supplement 2.** Temporal trend in total richness and abundance for warm-adapted species and cold-adapted species groups, seperately.

correlation between species' trends in colonization and extinction probabilities (Pearson correlation $r=-0.77$; **Figure 4b**).

Similar patterns were found for abundance, where we detected a strong positive relationship (t-test: slope = 0.153, $t_{58}=3.047$, p=0.003) between STI and each species' temporal trend in abundance: warm-adapted species generally increased abundance while cold-adapted species decreased over 10 years (**Figure 4c**). At the island scale, both total abundance and total species richness increased significantly for the warm-adapted species group while both responses significantly decreased for the

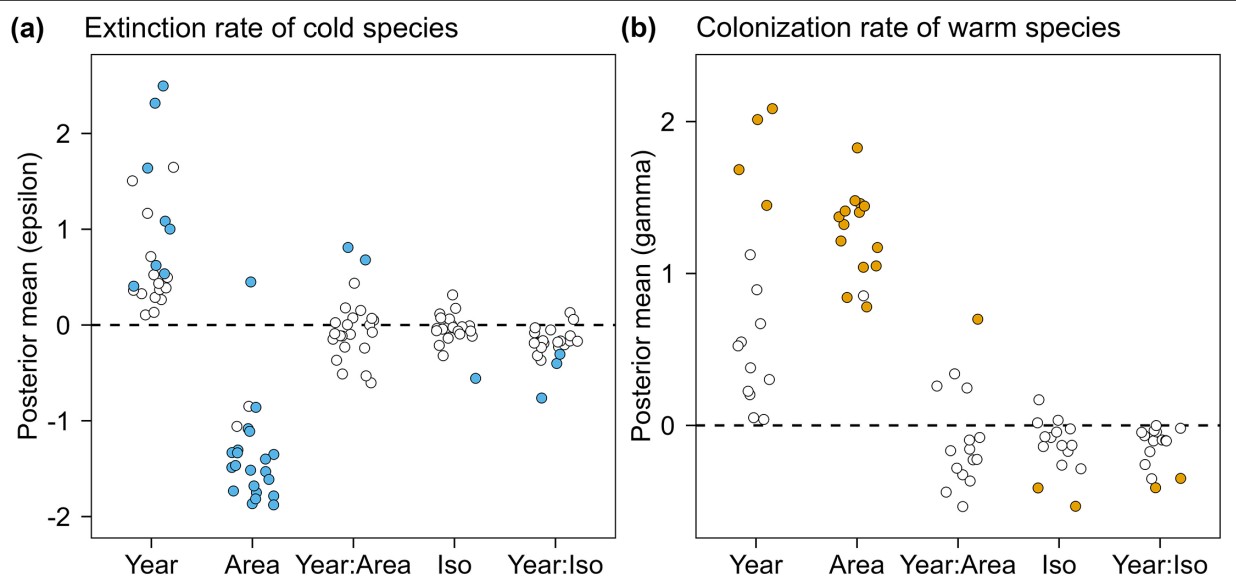

**Figure 5.** Posterior estimates of logit-scale parameters related to cold-adapted species' extinction rates and warm-adapted species' colonization rates. Points are species-specific posterior means on the logit-scale extracted from MSOM, where parameters >0 indicate positive effects (on extinction [a] or colonization [b]) and parameters <0 indicate negative effects. Only cold-adapted species (23 selected species) with positive trends in extinction (**a**) and warm-adapted species (15 selected species) with positive trends in colonization (**b**) are shown, which are two main processes contributing positively to thermophilization. Points in blue/orange indicate significant effects (95% or 80% credible intervals) and points in white indicate non-significant effects. All points were jittered slightly by 0.22 in width.

The online version of this article includes the following figure supplement(s) for figure 5:

**Figure supplement 1.** Relationship between island area and understory breeding season air temperature of 20 islands monitored in the year 2022.

cold-adapted species (richness trend, *P*<0.001; total abundance trend, p<0.001; *Figure 2—figure supplement 1*).

## The mediating effects of fragmentation on colonization and extinction processes

Both colonization and extinction probabilities showed evidence of being moderated by fragmentation (*Figure 5*). Of particular interest was how the temporal trend in dynamic parameters varied as a function of area or isolation (i.e., Year:Area or Year:Isolation effects). Although species-specific interaction terms were not generally significant for most species, overall trends could be assessed across species. In particular, the effect of isolation on temporal dynamics of thermophilization was relatively consistent across cold- (*Figure 5a*) and warm-adapted species (*Figure 5b*); specifically, on islands nearer to the mainland, warm-adapted species (15 out of 15 investigated species) increased their colonization probability at a higher rate over time, while most cold-adapted species (21 out of 23 species) increased their extinction probability at a higher rate.

Island area, in contrast, had a more mixed role in moderating colonization/extinction rates across species. For most warm-adapted species (11 out of 15 investigated species; *Figure 5b*), colonization rates increased faster over time on smaller islands (i.e., negative area:year interaction term); however, the colonization rates of the other four species increased faster on larger islands, including the only significant positive interactive effect (*Dendrocitta formosae*). For cold-adapted species, extinction rates were approximately equally split between increasing faster on smaller islands (12 out of 23 species) and increasing faster on larger islands (11 out of 23 species), including two species, *Spizixos semitorques* and *Pericrocotus cantonensis*, which showed significantly increasing rates of extinction on larger islands.

## Discussion

Our study explored the driving processes underlying a decade of thermophilization of a bird community breeding in a model fragmented system, specifically testing whether and how habitat fragmentation may mediate the colonization-extinction dynamics which underlie shifts in mean community traits. Our results confirm that community thermal niche composition was changing directionally, mainly through gains in warm-adapted species and individuals and losses in cold-adapted species and individuals. We further found that habitat fragmentation influences two processes of thermophilization: colonization rates of most warm-adapted species tended to increase faster on smaller and less isolated islands, while the loss rates of most cold-adapted species tended to be exacerbated on less isolated islands.

### Thermophilization of bird communities

Although thermophilization has been found in many taxa (*Lajeunesse and Fourcade, 2023*) and is more apparent in the tropics than in temperate zones (*Freeman et al., 2021*), there are few empirical demonstrations of this process in subtropical fragmented systems. We found decadal-scale thermophilization evident in both occurrence- and abundance-based metrics, although the response was considerably stronger for abundance. This result is consistent with work in other systems, where abundance-weighted CTI generally responds more quickly than occurrence-based CTI given climatic warming (*Devictor et al., 2008*; *Lindström et al., 2013*; *Oliver et al., 2017*; *Tayleur et al., 2016*). Notably, when tested on the landscape scale (versus on individual island communities), only the abundance-based thermophilization trend was significant, indicating thermophilization of bird communities was mostly due to occurrence dynamics within the region, rather than exogenous community turnover outside the region.

### Mechanisms underlying CTI trends

Consistent with our expectation, we found an overall increase in occupancy and abundance of warm-adapted species and a decrease in occupancy and abundance of cold-adapted species, which together contributed to thermophilization. This pattern is consistent with many previous studies conducted in continuous habitats in Europe or North America both for breeding and non-breeding bird communities (*Curley et al., 2022*; *Princé and Zuckerberg, 2015*; *Tayleur et al., 2016*). Also consistent with our expectation was that occupancy change was the product of temporal trends in both colonization and extinction dynamics; namely, warm-adapted species showed increasing colonization and decreasing extinction rates, while, cold-adapted species showed decreasing colonization and increasing extinction rates. Following that STI is predictably related to temporal trends in colonization and extinction rate (*Figure 4*), we can conclude that STI is an effective broad indicator of relative climatic vulnerability within a community, where species with relatively higher thermal preferences are generally poised to benefit from local climate warming and vice versa. This conclusion is supported by studies that have established relationships between STI and species' population trend (*Pearce-Higgins et al., 2015*; *Rigal et al., 2023*), species' contribution analysis (*Tayleur et al., 2016*), or spatial occupancy (*Anderson et al., 2023*). Decomposing average community trait shifts into colonization and extinction dynamics of individual species and their associated traits provides a comprehensive and more nuanced insight into community response to climate change.

There are numerous reasons why we would expect temperature to strongly drive community change, as temperature is related to species' survival through food resources, foraging time, and thermoregulatory requirements (*Dunn and Møller, 2019*). The local population growth of warm-adapted species may mirror their range expansion under climate warming. For example, thermophilization of avian communities in North America is mainly owed to the species that have increased in both range and population size (*Curley et al., 2022*). Contemporary latitudinal range shifts are thought to be primarily resultant of the expansion of polar margins of equatorial-distributed species (*La Sorte and Thompson, 2007*; *Thomas and Lennon, 1999*), possibly due to extinction lags at equatorial margins (*La Sorte and Jetz, 2012*). TIL is near the cold edge of most warm-adapted species like *Hemixos castanonotus*, *Pericrocotus solaris*, and *Lonchura punctulate*. Combined with the prediction that most birds in China should move upslope or poleward under climate change (*Hu et al., 2020*), we infer that local climate will continue to shift these warm-adapted species toward a thermal optimum, predicting continued population growth in the near future for these species (*Sagarin and Gaines, 2002*).

Meanwhile, most cold-adapted species in our system demonstrated increasing extinction rates over time. Extinctions are more difficult to detect than range expansions, partly due to temporal lags (*Thomas et al., 2006*; *Thomas and Lennon, 1999*) but also due to pervasive bias of false absences in time series data (*Tingley and Beissinger, 2009*). Moreover, species' extinctions are more likely to be mediated by factors such as rainfall, species interactions, and changes in vegetation (*Paquette and Hargreaves, 2021*; *Thomas and Lennon, 1999*), which will further lead to unequal rates of extinction relative to colonization. In our community, although there are many cold-adapted species – such as *Streptopelia orientalis*, *Phasianus colchicus*, *Passer montanus*, and *Emberiza cioides* – that have northern distributional limits much farther northern than our site, our site is not close to the current southern boundary for these species either. Even so, they still showed considerable loss in TIL, suggesting fragmented landscape exacerbated the negative effect of warming on these cold-adapted species.

Despite the general trend, not all warm-adapted species increased (in abundance or occupancy) and not all cold-adapted species decreased. In other words, not all species contributed equally to the process of thermophilization, as has been seen elsewhere (*Tayleur et al., 2016*). Variation in species' response relates to their true thermal tolerance in the study region, which would require additional physiological data to measure and could also be influenced by habitat usage and biotic interactions.

## The mediating effect of fragmentation on colonization and extinction processes

It has long been thought that habitat fragmentation can impact species' climate tracking (*Opdam and Wascher, 2004*) but there are only a few empirical studies to date (*Fourcade et al., 2021*; *Warren et al., 2001*). Our research examining colonization-extinction dynamics provides empirical evidence of the possible mechanisms in a land-bridge island system.

As expected, the mediating effect of isolation on the two main processes of thermophilization was relatively consistent, with warm-adapted species colonizing faster and cold-adapted species being extirpated faster on islands nearer to the mainland. This indicates that isolation from population sources can limit the dispersal that underlies range shifts, supporting the general perception that habitat fragmentation is thought to impede climate-driven range expansions (*Opdam and Wascher, 2004*) or even block them (*Warren et al., 2001*). While we cannot truly distinguish in our system between local extinction and emigration, we suspect that given two islands equal except in isolation, if both lose suitability due to climate change, individuals can easily emigrate from the island nearer to the mainland, while individuals on the more isolated island would be more likely to be trapped in place until the species went locally extinct due to a lack of rescue.

As a caveat, we only consider the distance to the nearest mainland as a measure of fragmentation, consistent with previous work in this system (*Si et al., 2014*), but we acknowledge that other distance-based metrics of isolation that incorporate inter-island connections and island size could hint on a more complex pattern going on in real-life than was assumed for this study, thus reveal additional insights on fragmentation effects. For instance, smaller islands may also potentially utilize species pools from nearby larger islands, rather than being limited solely to those from the mainland. The spatial arrangement of islands, like the arrangement of habitat, can influence niche tracking of species (*Fourcade et al., 2021*). Future studies should use a network approach to take these metrics into account to thoroughly understand the influence of isolation and spatial arrangement of patches in mediating the effect of climate warming on species.

Island size also had pervasive effects on underlying thermophilization dynamics. In our study, the colonization rate of a large proportion of warm-adapted species (11 out of 15) and half of cold-adapted species (12 out of 23) were increasing more rapidly on smaller islands. Similarly, a lower proportion of habitats enhances the colonization of warm-adapted species under climate change in the butterfly assemblage (*Fourcade et al., 2021*). The observed colonization-extinction patterns on smaller islands support the hypothesis that warm-adapted species benefit from the lowered thermal buffering of ambient temperature on smaller islands, while the same mechanism may speed up extinction (or emigration) of cold-adapted species (*Figure 1b*). To investigate this mechanistic hypothesis more thoroughly, using understory air temperature data monitored on 20 islands in the breeding season in 2022, we found that temperature buffering ability indeed was diminished on smaller islands

(*Figure 5—figure supplement 1*), further supporting microclimate as a primary driver of fragmentation's mediation of thermophilization (*Zellweger et al., 2020*).

The increased extinction rate of some cold-adapted species on smaller islands is also consistent with the hypothesized impacts of reduced habitat heterogeneity (*Figure 1c*). Previous studies found that areas with greater amounts of habitat featured lower extinction rates of cold-adapted butterfly species (*Fourcade et al., 2021*), possibly by facilitating movement to colder microclimates (*Suggitt et al., 2018*). We thus suppose that habitat heterogeneity could also mitigate the loss of these relatively cold-adapted species as expected. Habitat diversity, including the observed number of species, the rarefied species richness per island, species density and the rarefied species richness per unit area, all increased significantly with island area instead of isolation in our system (*Liu et al., 2020*). Thus, as habitat heterogeneity was only consistent with our observed response of cold-adapted species (*Figure 1c*), while microclimate buffering is consistent with observed responses of both warm- and cold-adapted species (*Figure 1b*), our evidence more consistently supports the latter than the former.

Contrary to any expected mechanisms (*Figure 1*), the extinction rate of some cold-adapted species – such as *Spizixos semitorques* and *Pericrocotus cantonensis* – was increased faster on larger islands. Although post hoc, we speculate that these species face stronger predation pressures on larger islands. For example, predators like snakes and felids may also increase activity under climate warming (*DeGregorio et al., 2015*), but these predators only exist in our system on the larger islands. Besides, *Garrulax canorus*, which is a warm-adapted species, increased faster on larger islands. This species may depend more on habitat heterogeneity than microclimate alone, including food resources and microrefugia, which is related to faster climate tracking (*Gaüzère et al., 2017*). Overall, these idiosyncratic responses reveal several possible mechanisms in regulating species' climate responses, including resource demands and biological interactions like competition and predation. Future studies are needed to take these factors into account to understand the complex mechanisms by which habitat loss meditates species range shifts.

Overall, our findings have important implications for conservation practices. Firstly, we confirmed the role of isolation in limiting range shifting. Better connected landscapes should be developed to remove dispersal constraints and facilitate species' relocation to the best suitable microclimate. Second, small patches can foster the establishment of newly adapted warm-adapted species while large patches can act as refugia for cold-adapted species. Therefore, preserving patches of diverse sizes can act as stepping stones or shelters in a warming climate depending on the thermal affinity of species. These insights are important supplement to the previous emphasis on the role of habitat diversity in fostering (*Richard et al., 2021*) or reducing (*Gaüzère et al., 2017*) community-level climate debt.

## Materials and methods
### Study area and islands selection
The Thousand Island Lake (TIL), located in eastern China, was formed in 1959 following the construction of the Xin'anjiang Dam for hydroelectricity (*Figure 2*). When the lake is at its highest, there are 1078 islands with an area larger than 0.25 ha. Currently, about 90% of the forested areas are dominated by Masson pine (*Pinus massoniana*) in the canopy and broad-leaved species (e.g. *Loropetalum chinensei*, *Vaccinium carlesii*, and *Rhododendron simsii*) in the sub-canopy and understory (*Liu et al., 2020*). The climatic zone is a typical monsoon climate. The precipitation is mainly concentrated between April and June with an average yearly rainfall of 1430 mm. The average annual temperature is 17 °C (hottest from June to August) and the average daily temperature ranges from –7.6 °C in January to 41.8 °C in July (*Si et al., 2024*).

We selected 36 islands according to a gradient of island area and isolation with a guarantee of no significant correlation between island area and isolation (Pearson $r$=–0.21, p=0.21). For each island, we calculated island area and isolation (measured in the nearest Euclidean distance to the mainland) to represent the degree of habitat fragmentation (*Figure 2*). Distance to the mainland is the best distance-based measure in fitting species' colonization rate and extinction rate in TIL (*Si et al., 2014*). Since lake formation, the islands have been protected by forbidding logging, allowing natural succession pathways to occur.

## Bird data

We established 53 survey transects on 36 islands with the sampling effort on each island roughly proportional to the logarithm of the island area (*Schoereder et al., 2004*). As a result, 53 transects on 36 islands were sampled, including eight transect trails on the largest study island (area = 1058 ha), four transects on two islands between 100 and 500 ha, two transects on four islands between 10 and 100 ha, and one on each of the remaining small islands (c. 1 ha for most islands) (*Si et al., 2018*). Breeding bird communities were surveyed on each transect nine times annually (three times per month from April to June) from 2012 to 2021 (*Si et al., 2017*). In each survey, observers walked along each transect at a constant speed (2.0 km/hr) and recorded all the birds seen or heard on the survey islands. To minimize the bias, the order and direction of each island surveyed were randomized. We based our abundance estimate on the maximum number of individuals recorded across the nine annual surveys. We excluded non-breeding species, nocturnal and crepuscular species, high-flying species passing over the islands (e.g., raptors, swallows) and strongly water-associated birds (e.g., cormorants) from our record. First, our surveys were conducted during the day, so some nocturnal and crepuscular species, such as the owls and nightjars were excluded because of inadequate survey design. Second, wagtail, kingfisher, and water birds such as ducks and herons were excluded because we were only interested in forest birds. Third, birds like swallows, and eagles who were usually flying or soaring in the air rather than staying on islands, were also excluded as it was difficult to determine their definite belonging islands. Following these filtering, 60 species were finally retained.

## Climate data

We obtained climate data (monthly average temperature from 2012 to 2021) from the Meteorological Bureau of Chun'an County in Zhejiang Province, China. We calculated breeding season temperature as the average of the mean monthly temperature from April to June in order to represent the thermal conditions birds experience at this site each year (*Lindström et al., 2013*). Breeding season temperature increased significantly over 10 years (slope = 0.078, $r^2$=34.53%, p=0.04, *Figure 2—figure supplement 1*).

## STI and CTI

We followed the methods of *Devictor et al., 2008* to calculate a species temperature index (STI) using ArcMap 10.3. STI of a given species is defined as the average temperature of breeding months (April – June) across its distributional range (restricted to the Northern hemisphere) averaged over 1970–2000. Monthly temperature data were obtained from WorldClim at a resolution of 30-arc seconds (http://www.worldclim.org). Distributional maps were extracted from Birdlife International 2019 (http://datazone.birdlife.org/species/requestdis), in which we selected only the distributional regions where the species is either resident or breeding. All species were divided into two groups indicating their relative thermal preference: species with STI higher than the median STI were labeled as warm-adapted species, and species lower than the median STI were labeled as cold-adapted species (*Bates et al., 2017*). The STI of a species can differ depending on distributional and climate data but the rank order of species STI is highly correlated across methods (*Barnagaud et al., 2013*). We verified the robustness of our STI calculations using different distributional ranges and annual time windows (*Figure 3—figure supplement 2*).

CTI is a community-level index representing the average species thermal niche of all species or individuals in the community (*Devictor et al., 2008*). Accordingly, for each community in each year, CTI was calculated as the average STI of all occurring species (hereafter: CTI$_{occur}$) or counted individuals (hereafter: CTI$_{abun}$):

$$\text{CTI}_{\text{occur}, j, t} = \frac{\sum_{i=1}^{N_{j,t}} STI_i}{N_{j,t}}$$

$$\text{CTI}_{\text{abun}, j, t} = \frac{\sum_{i=1}^{N_{j,t}} \left( STI_i \times A_{i,j,t} \right)}{\sum_{i=1}^{N_{t,j}} A_{i,j,t}}$$

where N$_{j,t}$ is the total number of species surveyed in the community *j* in year *t*, A$_{i,j,t}$ denoted the maximum abundance among nine surveys of the *i*th species in community *j* in year *t*.

$CTI_{occur}$ was calculated using occurrence data corrected for imperfect detection (see next section), while $CTI_{abun}$ was estimated using the maximum annual count across 9 surveys. Given that our survey effort was so high (i.e., 9 repeat surveys per year), true abundance should be highly correlated with maximum observed abundance (**MacKenzie et al., 2006**). We also used these occurrence and abundance metrics from all islands to compute each year's regional CTI at the landscape level.

## Multispecies dynamic occupancy model

Presence-absence data can be highly sensitive to false negatives, so to explore the driving processes of change in $CTI_{occur}$, and to explore how habitat fragmentation mediated the main processes while accounting for the imperfect detection, we developed a spatially hierarchical dynamic multi-species occupancy model (MSOM) in a Bayesian framework based on the model by **Royle and Kéry, 2007**. We denote $y_{nijt}$ as the observation (detected = 1; undetected = 0) for species $n$ (1–60 species) in survey visit $j$ (1–9 surveys) at transect $i$ (1–53 transects) in year $t$ (1–10 years). Observation, $y_{nijt}$ is assumed to be the result of imperfect detection of the true occurrence status, $z_{nit}$ (1 or 0), and is thus modeled as a Bernoulli-distributed variable with a probability of $p_{nijt} \times z_{nit}$, where $p_{nijt}$ is the probability of detection for a given survey along a transect:

$$y_{nijt} \sim Bernoulli\left(z_{nit} * p_{nijt}\right)$$

where $z_{nit}$ is assumed to be of fixed presence/absence status for a given species across all $j$ survey intervals within year $t$. However, given that individual birds move dynamically across territories within a year, during 9 surveys each year, we broadly interpret our estimates of 'occupancy' as 'use' (**Si et al., 2018**) to relax the closure assumption (**MacKenzie et al., 2004**).

Each site has an initial value of occupancy ($\psi_{ni1}$) which can be given a uniform prior distribution from 0 to 1, and occupancy ($\psi_{nit}$) in subsequent years ($t>1$) is determined based on whether sites become colonized or remain occupied through persistence **MacKenzie et al., 2003**:

$$Z_{nit} \sim Bernoulli\left(\psi_{nit}\right)$$
$$\psi_{nit} = Z_{ni,t-1}\left(1 - \varepsilon_{ni,t-1}\right) + \left(1 - Z_{ni,t-1}\right)\gamma_{ni,t-1} \text{ for } t > 1$$

where $\psi_{nit}, \varepsilon_{ni,t-1}, \gamma_{ni,t-1}$ are transect-level probabilities of occupancy, extinction, and colonization, respectively.

We modeled the probability of detection, $p_{nijt}$, as a function of the length of the transect, $length_i$ and the ordinal day of year, $day_{ijt}$. Island identity was included as a random effect to account for the nonindependence of transects within the same islands.

$$logit\left(p_{nijt}\right) = p0_n + p1_n length_i + p2_n day_{ijt} + island_{REni}$$

To explore how colonization rate and extinction rate change across 10 years and how habitat fragmentation mediated these dynamics processes, we modeled site-level extinction and colonization ($\varepsilon_{ni,t}, \gamma_{ni,t}$) each as a logit-linear function of five covariates: year, island area, isolation, the interaction between year and area and between isolation and year. Year was added as a random slope, thus allowing the temporal trend in the colonization or extinction to vary with island area and isolation:

$$logit(\varepsilon_{ni,t}) = \alpha0_n + \alpha1_n area_i + \alpha2_n isolation_i + (\alpha3_n + RE_i)year_t + \alpha4_n area_i year_t$$
$$+\alpha5_n isolation_i year_t + island_{REni}$$
$$logit(\gamma_{ni,t}) = \beta0_n + \beta1_n area_i + \beta2_n isolation_i + (\beta3_n + RE_i)year_t + \beta4_n area_i year_t$$
$$+\beta5_n isolation_i year_t + island_{REni}$$

In all cases, continuous covariates were z-transformed (mean of 0 and a standard deviation of 1). There is no strong correlation between island area and isolation (Pearson correlation $r=-0.214$) so these effects can be considered independent. We fit the MSOM with JAGS (**Plummer, 2003**) using the package rjags (**Plummer, 2023**) in R v4.3.1 (**R Development Core Team, 2023**). We used vague priors (e.g., normal with $\mu=0$, $\tau=0.01$). We ran three chains for 60,000 iterations, discarded the first 40,000 as burn-in and thinned by 20, yielding a combined posterior sample of 3000. We extracted 600 iterations across three chains for Z to calculate $CTI_{occur}$. Convergence was checked visually with trace plots and

confirmed with a Gelman–Rubin statistic <1.1 (***Gelman and Rubin, 1992***). Inference on parameters was made using 95% and 80% Bayesian credible intervals. We checked the posterior predictive ability of the model fit by calculating Bayesian p-values (p=0.488, indicating an unbiased estimation of our model; ***Gelman et al., 1996***).

## Statistical analysis

### Thermophilization of bird communities

To test for occurrence-based thermophilization at the island level, Bayesian linear regression using R2jags (***Su and Yajima, 2021***) was used to derive a posterior estimate of the temporal trend in $CTI_{occur}$ while propagating error in the estimation of $CTI_{occur}$. The continuous variable *year* (z-transformed) was incorporated as the only predictor. To account for the nonindependence of data within islands, we included island identity as a random intercept, thus allowing intercepts to vary across islands. We used vague priors (e.g., normal with μ=0, $\tau$=0.001) for intercept and year effect. We ran three chains for 10,000 iterations, discarded the first 5000 as burn-in and thinned by 5, yielding a combined posterior sample of 3000. We used the function *MCMCsummary* to get posterior intervals of the intercept and year effect, which were then used for plotting.

To test for abundance-based thermophilization, linear mixed effect models (LMM) with restricted maximum likelihood (REML) were used to model temporal trends in $CTI_{abun}$ using function *lme* in R package nlme (***Pinheiro et al., 2017***). As with the occurrence analysis, we used the continuous variable *year* (z-transformed) as the only fixed effect and island as a random intercept. To account for the temporal autocorrelation of CTI from the same island through time, we included the first-order correlation structure (e.g. the corAR1 structure) because observations sampled from the nearby years may be more similar.

We also tested whether the overall regional bird composition (i.e. gamma diversity) experienced thermophilization. Bayesian linear regression was used for modeling temporal trends in regional $CTI_{occur}$. For abundance, a generalized least square (GLS) model was used to test the relationship between $CTI_{abun}$ and continuous variable *year* while accounting for potential temporal autocorrelation. GLS was conducted using *gls* function in nlme R package (***Pinheiro et al., 2023***).

### Mechanisms underlying CTI trends

To explore the dynamics underlying trends in $CTI_{occur}$, specifically, whether thermophilization was accompanied by increasing occupancy of warm-adapted species and decreasing occupancy of cold-adapted species, we extracted 600 posterior samples of occupancy rate per species, and summarized $\psi_{nt}$ by its mean and variation (logit-transformed to meet normality). We then modeled each species' temporal trend in occupancy ($\psi_{nt}$) over time using Bayesian linear regression in R2jags (***Su and Yajima, 2021***) to account for propagated uncertainty in $\psi_{nt}$. Similarly, to decompose the $CTI_{abun}$ trend, we modeled each species' temporal trend in maximum abundance using a generalized linear mixed effect model with Poisson error structure while accounting for potential overdispersion – via an observation-specific random effect (***Harrison, 2014***). Weighted linear regression models tested for a relationship between STI and occupancy trend and between STI and abundance trend. This analysis was conducted using function *lmer* in package lme4 (***Bates et al., 2015***). Temporal correlations between occupancy and colonization or extinction rates were measured with Pearson correlations, using colonization and extinction rates derived directly from the fitted MSOM.

As a complementary analysis, we also investigated the trend in species richness or total abundance of the two thermal preference groups, separately. The total richness or abundance in each group was calculated on each transect and in each year. We then modeled temporal trends of natural logarithm-transformed richness or abundance using LMM with island identity as the random effect. Partial regression plots from LMM were produced using ggeffects R package (***Lüdecke, 2018***). For each model, we checked for normality of residuals and computed goodness-of-fit metrics ($R^2$) including conditional and marginal $R^2$ (***Nakagawa and Schielzeth, 2013***) using performance R package (***Lüdecke et al., 2021***). All continuous fixed effects in the models were standardized (mean = 0 and SD = 1) to ease computation and facilitate the interpretation and comparison of coefficients within and across models (***Schielzeth, 2010***).

## The mediating effects of habitat fragmentation on colonization and extinction processes

An increasing colonization trend of warm-adapted species and increasing extinction trend of cold-adapted species are two main expected processes that cause thermophilization (*Fourcade et al., 2021*). To test our third prediction about the mediating effect of habitat fragmentation, we selected warm-adapted species that had an increasing trend in colonization rate (positive year effect in colonization rate) and cold-adapted species that had an increasing extinction rate (positive year effect in extinction rate). For each of these species, we extracted their posterior effect of year, area, isolation, and the interaction terms from previously fit models. The interaction terms reveal if habitat fragmentation positively or negatively affects either colonization or extinction processes.

## Acknowledgements

We sincerely thank numerous graduate students in our group for the bird surveys in the field, and the Xin'an River Ecological Development Group Corporation, Chun'an Forestry Bureau, and the Thousand Island Lake National Forest Park for research permits. We also thank Dr. Di Zeng and Dr. Yuhao Zhao from East China Normal University for their insightful comments on the research. Zhe Liu from Ludong University helped a lot in learning ArcMap. This study is supported by the National Natural Science Foundation of China (#32030066 to Ping Ding.; #32071545 to Xingfeng Si), Natural Science Foundation of Zhejiang Province (#LD21C030002 to Ping Ding).

## Additional information

### Funding

| Funder | Grant reference number | Author |
| --- | --- | --- |
| National Natural Science Foundation of China | #32030066 | Ping Ding |
| National Natural Science Foundation of China | #32071545 | Xingfeng Si |
| Natural Science Foundation of Zhejiang Province | #LD21C030002 | Ping Ding |

The funders had no role in study design, data collection and interpretation, or the decision to submit the work for publication.

### Author contributions

Juan Liu, Conceptualization, Data curation, Formal analysis, Visualization, Methodology, Writing - original draft, Writing - review and editing; Morgan W Tingley, Formal analysis, Visualization, Methodology, Writing - review and editing; Qiang Wu, Peng Ren, Data curation, Visualization, Writing - original draft, Writing - review and editing; Tinghao Jin, Data curation, Formal analysis, Visualization, Methodology, Writing - original draft, Writing - review and editing; Ping Ding, Xingfeng Si, Conceptualization, Data curation, Supervision, Funding acquisition, Methodology, Writing - review and editing

### Author ORCIDs

Juan Liu https://orcid.org/0000-0001-7343-7877
Morgan W Tingley https://orcid.org/0000-0002-1477-2218
Peng Ren https://orcid.org/0000-0001-6033-6188
Tinghao Jin http://orcid.org/0000-0002-7088-9963
Xingfeng Si https://orcid.org/0000-0003-4465-2759

Reviewer #3 (Public review): https://doi.org/10.7554/eLife.98056.4.sa1
Author response https://doi.org/10.7554/eLife.98056.4.sa2

# Additional files

## Supplementary files
• MDAR checklist

## Data availability
The data and R code that support the findings of this study are openly available in figshare at https://doi.org/10.6084/m9.figshare.25132442.v1.

The following dataset was generated:

| Author(s) | Year | Dataset title | Dataset URL | Database and Identifier |
|-----------|------|---------------|-------------|-------------------------|
| Liu J, Tingley MW, Wu Q, Ren P, Jin T, Ding P, Si X | 2024 | Habitat fragmentation mediates the mechanisms underlying long-term climate-driven thermophilization in birds | https://doi.org/10.6084/m9.figshare.25132442 | figshare, 10.6084/m9.figshare.25132442.v1 |

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
