## [Editor Report · eLife Assessment]

This **fundamental** study substantially advances our understanding of how habitat fragmentation and climate change jointly influence bird community thermophilization in a fragmented island system. The authors provide **convincing** evidence using appropriate and validated methodologies to examine how island area and isolation affect the colonization of warm-adapted species and the extinction of cold-adapted species. This study is of high interest to ecologists and conservation biologists, as it provides insight into how ecosystems and communities respond to climate change.

---

## [Referee Report · Reviewer #3 (Public review)]

Summary:

Juan Liu et al. investigated the interplay between habitat fragmentation and climate-driven thermophilization in birds in an island system in China. They used extensive bird monitoring data (9 surveys per year per island) across 36 islands of varying size and isolation from the mainland covering 10 years. The authors use extensive modeling frameworks to test a general increase of the occurrence and abundance of warm-dwelling species and vice versa for cold-dwelling species using the widely used Community Temperature Index (CTI), as well the relationship between island fragmentation in terms of island area and isolation from the mainland on extinction and colonization rates of cold- and warm-adapted species. They found that indeed there was thermophilization happening during the last 10 years, which was more pronounced for the CTI based on abundances and less clearly for the occurrence based metric. Generally, the authors show that this is driven by an increased colonization rate of warm-dwelling and an increased extinction rate of cold-dwelling species. Interestingly, they unravel some of the mechanisms behind this dynamic by showing that warm-adapted species increased while cold-dwelling decreased more strongly on smaller islands, which is - according to the authors - due to lowered thermal buffering on smaller islands (which was supported by air temperature monitoring done during the study period on small and large islands). They argue, that the increased extinction rate of cold-adapted species could also be due to lowered habitat heterogeneity on smaller islands. With regards to island isolation, they show that also both thermophilization processes (increase of warm and decrease of cold-adapted species) was stronger on islands closer to the mainland, due to closer sources to species populations of either group on the mainland as compared to limited dispersal (i.e. range shift potential) in more isolated islands.

The conclusions drawn in this study are sound, and mostly well supported by the results. Only few aspects leave open questions and could quite likely be further supported by the authors themselves thanks to their apparent extensive understanding of the study system.

Strengths:

The study questions and hypotheses are very well aligned with the methods used, ranging from field surveys to extensive modeling frameworks, as well as with the conclusions drawn from the results. The study addresses a complex question on the interplay between habitat fragmentation and climate-driven thermophilization which can naturally be affected by a multitude of additional factors than the ones included here. Nevertheless, the authors use a well balanced method of simplifying this to the most important factors in question (CTI change, extinction, colonization, together with habitat fragmentation metrics of isolation and island area). The interpretation of the results presents interesting mechanisms without being too bold on their findings and by providing important links to the existing literature as well as to additional data and analyses presented in the appendix.

Weaknesses:

The metric of island isolation based on distance to the mainland seems a bit too oversimplified as in real-life the study system rather represents an island network where the islands of different sizes are in varying distances to each other, such that smaller islands can potentially draw from the species pools from near-by larger islands too - rather than just from the mainland. Although the authors do explain the reason for this metric, backed up by earlier research, a network approach could be worthwhile exploring in future research done in this system. The fact, that the authors did find a signal of island isolation does support their method, but the variation in responses to this metric could hint on a more complex pattern going on in real-life than was assumed for this study.

Comments on revisions:

I'm happy with the revisions made by the authors.

---

## [Author Response]

The following is the authors’ response to the previous reviews.

**Public Reviews:**

**Reviewer #3 (Public review):**
Summary:Juan Liu et al. investigated the interplay between habitat fragmentation and climate-driven thermophilization in birds in an island system in China. They used extensive bird monitoring data (9 surveys per year per island) across 36 islands of varying size and isolation from the mainland covering 10 years. The authors use extensive modeling frameworks to test a general increase of the occurrence and abundance of warm-dwelling species and vice versa for cold-dwelling species using the widely used Community Temperature Index (CTI), as well the relationship between island fragmentation in terms of island area and isolation from the mainland on extinction and colonization rates of cold- and warm-adapted species. They found that indeed there was thermophilization happening during the last 10 years, which was more pronounced for the CTI based on abundances and less clearly for the occurrence based metric. Generally, the authors show that this is driven by an increased colonization rate of warm-dwelling and an increased extinction rate of cold-dwelling species. Interestingly, they unravel some of the mechanisms behind this dynamic by showing that warm-adapted species increased while cold-dwelling decreased more strongly on smaller islands, which is - according to the authors - due to lowered thermal buffering on smaller islands (which was supported by air temperature monitoring done during the study period on small and large islands). They argue, that the increased extinction rate of cold-adapted species could also be due to lowered habitat heterogeneity on smaller islands. With regards to island isolation, they show that also both thermophilization processes (increase of warm and decrease of cold-adapted species) was stronger on islands closer to the mainland, due to closer sources to species populations of either group on the mainland as compared to limited dispersal (i.e. range shift potential) in more isolated islands.The conclusions drawn in this study are sound, and mostly well supported by the results. Only few aspects leave open questions and could quite likely be further supported by the authors themselves thanks to their apparent extensive understanding of the study system.Strengths:The study questions and hypotheses are very well aligned with the methods used, ranging from field surveys to extensive modeling frameworks, as well as with the conclusions drawn from the results. The study addresses a complex question on the interplay between habitat fragmentation and climate-driven thermophilization which can naturally be affected by a multitude of additional factors than the ones included here. Nevertheless, the authors use a well balanced method of simplifying this to the most important factors in question (CTI change, extinction, colonization, together with habitat fragmentation metrics of isolation and island area). The interpretation of the results presents interesting mechanisms without being too bold on their findings and by providing important links to the existing literature as well as to additional data and analyses presented in the appendix.Weaknesses:The metric of island isolation based on distance to the mainland seems a bit too oversimplified as in real-life the study system rather represents an island network where the islands of different sizes are in varying distances to each other, such that smaller islands can potentially draw from the species pools from near-by larger islands too - rather than just from the mainland. Although the authors do explain the reason for this metric, backed up by earlier research, a network approach could be worthwhile exploring in future research done in this system. The fact, that the authors did find a signal of island isolation does support their method, but the variation in responses to this metric could hint on a more complex pattern going on in real-life than was assumed for this study.

Thank you again for this suggestion. Based on the previous revision, we discussed more about the importance of taking the island network into future research. The paragraph is now on Lines 294-304:

“As a caveat, we only consider the distance to the nearest mainland as a measure of fragmentation, consistent with previous work in this system (Si et al., 2014), but we acknowledge that other distance-based metrics of isolation that incorporate inter-island connections and island size could hint on a more complex pattern going on in real-life than was assumed for this study, thus reveal additional insights on fragmentation effects. For instance, smaller islands may also potentially utilize species pools from nearby larger islands, rather than being limited solely to those from the mainland. The spatial arrangement of islands, like the arrangement of habitat, can influence niche tracking of species (Fourcade et al., 2021). Future studies should use a network approach to take these metrics into account to thoroughly understand the influence of isolation and spatial arrangement of patches in mediating the effect of climate warming on species.”

**Recommendations for the authors:**

**Reviewer #3 (Recommendations for the authors):**
Great job on the revision! The new version reads well and in my opinion all comments were addressed appropriately. A few additional comments are as follows:

Thank you very much for your further review and recognition. We have carefully modified the manuscript according to all recommendations.

(1) L 62: replace shifts with process

Done. We also added the word “transforming” to match this revision. The new sentence is now on Lines 61-63:

“Habitat fragmentation, usually defined as the process of transforming continuous habitat into spatially isolated and small patches”

(2) L 363: Your metric for habitat fragmentation is isolation and habitat area and I think this could be introduced already in the introduction, where you somewhat define fragmentation (although it could be clearer still). You could also discuss this in the discussion more, that other measures of fragmentation may be interesting to look at.

Thank you for this suggestion. We now introduced metric of habitat fragmentation in the Introduction part after habitat fragmentation was defined. The sentence is now on Lines 64-66:

“Among the various ways in which habitat fragmentation is conceptualized and measured, patch area and isolation are two of the most used measures (Fahrig, 2003).”

(3) L 384: replace for with because of

Done.

(4) L 388: "Following this filtering, 60 ...."

Done.

(5) Figure 1: In panels b-d you use different terms (fragmented, small, isolated) but aiming to describe the same thing. I would highly recommend to either use fragmented islands or isolated islands for all panels. Although I see that in your study fragmentation includes both, habitat loss and isolation. So make this clear in the figure caption too...

Thank you very much for this suggestion. It’s important to maintain consistency in using “fragmentation”. We change “fragmented, small, isolated” into “Fragmented patches” in the caption of b-d. The modified caption is now on Line 771:

(6) L 783: replace background with habitat (or landscape) and exhibit with exemplify

Done. The new sentence is now on Lines 782-784:

“The three distinct patches signify a fragmented landscape and the community in the middle of the three patches was selected to exemplify colonization-extinction dynamics in fragmented habitats.”

(7) One bigger thing is the definition of fragmentation in your study for which you used habitat area (from habitat loss process) and isolation. This could still be clarified a bit more, especially in the figures. In Fig. 1 the smaller panels b-d could all be titled fragmented islands as this is what the different terms describe in your study (small, isolated) and thus the figure would become even clearer. Otherwise I'm happy with the changes made.

Thank you for raising this important question. Yes, “habitat fragmentation” in our research includes both habitat loss and fragmentation per se. We have clarified the caption of b-d in Figure 1 as suggested by Recommendation (5). We believe this can make it clearer to the readers.